# The Role of Genomics and Proteomics in Lung Cancer Early Detection and Treatment [note 1]

**DOI:** 10.3390/cancers14205144

**Published:** 2022-10-20

**Authors:** Mohammad Hadi Abbasian, Ali M. Ardekani, Navid Sobhani, Raheleh Roudi

**Affiliations:** 1Department of Medical Genetics, National Institute of Genetic Engineering and Biotechnology (NIGEB), Tehran 1497716316, Iran; 2Department of Medical Biotechnology, National Institute of Genetic Engineering and Biotechnology (NIGEB), Tehran 1497716316, Iran; 3Department of Medicine, Section of Epidemiology and Population Sciences, Baylor College of Medicine, Houston, TX 77030, USA; 4Department of Radiology, Molecular Imaging Program at Stanford, Stanford University, Stanford, CA 94305, USA

**Keywords:** lung cancer, genomic markers, epigenomes markers, immunotherapy

## Abstract

**Simple Summary:**

Biomarkers for the prediction and efficacy of therapies are an urgent necessity for lung cancer patients. In this article, we summarize genomic and proteomic biomarkers utilized for the early detection and treatment of lung cancer, with a focus on immune checkpoint and PI3K pathways.

**Abstract:**

Lung cancer is the leading cause of cancer-related death worldwide, with non-small-cell lung cancer (NSCLC) being the primary type. Unfortunately, it is often diagnosed at advanced stages, when therapy leaves patients with a dismal prognosis. Despite the advances in genomics and proteomics in the past decade, leading to progress in developing tools for early diagnosis, targeted therapies have shown promising results; however, the 5-year survival of NSCLC patients is only about 15%. Low-dose computed tomography or chest X-ray are the main types of screening tools. Lung cancer patients without specific, actionable mutations are currently treated with conventional therapies, such as platinum-based chemotherapy; however, resistances and relapses often occur in these patients. More noninvasive, inexpensive, and safer diagnostic methods based on novel biomarkers for NSCLC are of paramount importance. In the current review, we summarize genomic and proteomic biomarkers utilized for the early detection and treatment of NSCLC. We further discuss future opportunities to improve biomarkers for early detection and the effective treatment of NSCLC.

## 1. Introduction

Worldwide, the second most common type of cancer diagnosed in men and women is lung cancer, which is the most common cause of cancer-related deaths. There are two main types of lung cancers: non-small-cell lung cancer (NSCLC) and small-cell lung cancer (SCLC). Adenocarcinoma and squamous cell carcinoma are the main histological subtypes of NSCLC. Prostate, colon, and breast cancers have screening tools for early detection; however, there are generally less accurate early detection biomarkers for lung cancer [1]. GLOBOCAN estimated approximately 2.2 million new lung cancer cases and 1,796,144 deaths [2].

The five-year survival rate in stage I NSCLC patients is 70–90%. However, over 75% of lung cancer patients are diagnosed at advanced or metastatic stages when the five-year survival rates are significantly lower [2,3]. Its diagnosis at advanced stages is associated with significantly lower five-year survival.

The first strategies for lung cancer detection for many years had been annual chest radiography and sputum cytology. It has not been until recently, as medicine has advanced, that these strategies have been updated. The results of several studies, such as two studies conducted by Johns Hopkins University and the Memorial Sloan Kettering Center, revealed that chest X-ray (CXR) and sputum cytology had not been promising screening tools for reducing lung cancer mortality [4,5]. The Early Lung Cancer Action Project (ELCAP) was initiated in 1993 and evaluated high-risk persons for lung cancer by low-radiation-dose computed tomography (low-dose C.T.) [6]. In addition, the National Lung Screening Trial (NLST) was initiated in 2002. These studies showed that the use of low-dose C.T. reduced mortality from lung cancer; however, cumulative radiation exposure, a high-false positive rate, and costs are some disadvantages of low-dose C.T. [7]. More specific, less invasive, and cost-effective biomarkers are therefore urgently needed.

In the past several decades, the molecular profiling of lung cancer patients has shown that many mutations correlate with NSCLC [8,9,10]. Molecular profiling helps stratify patients with specific mutations to use several lines of targeted therapies. *EGFR*, *KRAS*, *MET*, *BRAF*, *ERBB2*, *ALK*, *ROS1*, and *RET* gene rearrangements are the main actionable genetic alterations in NSCLC [11,12,13,14]. Several generations of agents have been developed to target these mutations, improving the patients’ outcomes. However, disease progression and resistance to targeted drugs remain major challenges in NSCLC cancer treatment. In our opinion, the efficacy of these drugs depends on how accurate biomarkers could help stratify patients that would best respond to such targeted therapies. As medicine advances, more accurate biomarkers have been discovered that could better help predict therapy responses. Proteins are functional products from the genome. The dynamic range of protein concentration differs from cell to cell and from time to time. Therefore, proteomics has dynamic and complex entities. A quantitative understanding of proteomics provides the exact knowledge about protein networks of human cells and may improve our cancer detection and treatment tools [15].

The application of proteomics has resulted in subtype-specific, stage-specific, and metastasis-specific tumor biomarkers in lung cancer. Numerous studies have demonstrated that a significant number of proteins, including ENO1, selenium-binding protein 1 (SELENBP1), carbonic anhydrase (CA), heat shock 20KD-like protein, and transgelin (SM22-alpha), are associated with poor prognosis [16,17].

This review summarizes the current knowledge on genomics and proteomics biomarkers for the early detection and treatment of NSCLC. Furthermore, we focus on new biomarkers predicting the efficacy of immunotherapies and the PI3K–mTOR–AKT axis. Finally, we summarize the ongoing clinical trials investigating the AKT–PI3K–mTOR pathway in NSCLC.

## 2. Early Detection

Cancer early detection increases the opportunity for more successful treatment. Five years of survival in NSCLC patients are approximately 63% and 7% for localized and distant tumors, respectively [18]. Early detection of lung cancer will dramatically reduce the mortality rate of patients. Many genomic and proteomic biomarkers, such as DNA methylation, microRNA (miRNA), antitumor antibodies, and plasma proteins, have been investigated for early cancer detection, including lung cancer. Liquid biopsy comprises various sources, including sputum, blood, exhaled breath, sputum, bronchial aspirate, and bronchoalveolar lavages for the early detection of lung cancer [19,20,21] (Figure 1).

### 2.1. Genomic and Epigenomes

#### 2.1.1. Methylation

DNA methylation (hypermethylation or hypomethylation) is an epigenetic modification that plays a pivotal role in silencing gene transcription and is crucial in many biological processes, including embryonic development, X-chromosome inactivation, genomic imprinting, chromatin structure, normal growth, and cellular proliferation. Abnormal DNA methylation has been observed in tumor initiation and proliferation of all forms of cancers [22]. The epigenetic pathway is potentially reversible and is involved in tumorigenesis by three main mechanisms: DNA hypermethylation, global hypomethylation of the genome, and histone modifications.

DNA methylation can be assessed by different techniques, such as digital PCR (dPCR), methylation-specific PCR (MSP), methyl-CpG-binding domain (MBD), real-time quantitative MSP, quantitative methylation-specific PCR (qMSP), multiplex nested methylation-specific PCR, and whole-genome bisulfite sequencing (WGBS) [23]. Aberrant DNA methylation is catalyzed by DNA methyltransferases (DNMTs) through the promoter methylation of tumor suppressor genes. It is associated with many types of cancers, including NSCLC, autoimmune diseases, diabetes, and multiple sclerosis [24,25,26]. Hypermethylation of DNA can silence tumor suppressor genes, and hypomethylation of DNA can lead to transcriptional activation proto-oncogenes. Methylation patterns have been associated with different cancer aspects, such as disease stage, survival, and response to treatment. DNA methylation could predict the response to platinum-based chemotherapy in lung cancer patients [27,28].

Methylation of single or panel genes is reported in lung cancer and is associated with response to therapies (Table 1). Several studies have shown that the methylation of *RASSF1A*, *SHOX2*, and *PTGER4* genes is greatly helpful in the early diagnosis of lung cancer [29,30,31]. *RASSF1A* is a tumor-suppressor gene, and its methylation-associated inactivation is reported in NSCLC, breast, and gastrointestinal cancer (G.C.) [32]. Combining two or more gene methylation detection methods have improved sensitivity and specificity compared to a single-gene methylation detection method. The promoter methylation *TAC1*, *HOXA17*, and *SOX17* in sputum obtained from a prospective cohort of 150 NSCLC patients and 60 controls showed sensitivity and specificity of 98% and 71%, respectively [33]. The sensitivity and specificity of the methylation panel of SOX17, *HOXA9*, *AJAP1*, *PTGDR*, *UNCX*, and *MARCH11* in NSCLC patients were 96.7% and 60%, respectively [34]. Based on the study by Liu et al., the methylation frequency *PCDHGB6*, *HOXA9*, *MGMT*, and *miR-126* reached 85.2% sensitivity and 81.5% specificity [34,35].

#### 2.1.2. miRNAs

MicroRNAs (miRNAs) are a large family of short single-stranded RNAs that regulate gene expression. MicroRNAs play a pivotal role in biological processes such as cell proliferation, differentiation, and survival [44]. In cancer, miRNAs may act as either tumor suppressors or oncogenes [45,46]. Profiling miRNA in plasma, serum, and sputum are promising noninvasive biomarkers for the early detection of NSCLC (Table 2). Numerous studies have reported the great potential of miRNAs in lung cancer diagnosis. A recent systematic review and meta-analysis by Wang et al. declared that miRNA-21 has potential clinical value in the diagnosis and prognosis of lung cancer [47]. Lu et al. investigated 723 human microRNAs in 106 plasmas in healthy individuals and patients with NSCLC or SCLC. They identified the diagnostic features of six miRNAs (miR-17, miR-190b, miR-19a, miR-19b, miR-26b, and miR-375) that have value for discriminating lung cancer patients from healthy individuals [48]. A panel of three miRNAs (miR-125a-5p, miR-25, and miR-126) in the study of Wang et al. distinguished early stage lung cancer patients from control subjects [49]. Pan et al. studied the expression of miR-33a-5p and miR-128-3p lung cancer tissues and cell lines. They found that the expression of these miRNAs was downregulated in tissues, cell lines, and the whole blood of early stage lung cancer patients [50]. A multicenter study analyzed bloodborne RNA signatures in 3102 patients and found that a 15-miRNA signature might help distinguish patients diagnosed with lung cancer from cancer-free individuals [51]. Khandelwal et al. explored the role of miR-590-5p in NSCLC patients in plasma samples, and they found that miRNA functions as a tumor suppressor in NSCLC and downregulated in NSCLC patients compared to healthy controls [52].

### 2.2. Proteomic Early Detection

Proteomics is a powerful approach to studying numerous proteins that generates information about biomarker identification, molecular interactions, and signaling pathways [60]. Mass spectrometry (MS) is an analytical tool in biomarker discovery that can be used to measure the mass-to-charge ratio of particles. MS-based proteomics has been applied to biomarker discovery for several decades. In oncology, MS approaches provided a decisive result for the early detection of breast, colorectal, prostate, and ovarian cancer [61,62,63,64]. Proteomic biomarkers could be used from several sources, including plasma, serum, tears, sputum, saliva, and urine [65,66,67]. We can also find proteomic biomarkers in exhaled breath condensate, bronchoalveolar lavage fluid, and pleural effusion for lung cancer [68] (Table 3). Zhang et al. profiled urine proteome in several cancer types and healthy control. In this study, five biomarkers (FTL: ferritin light chain; MAPK1IP1L: mitogen-activated protein kinase 1 interacting protein 1-like; FGB: fibrinogen beta chain; RAB33B: RAB33B, member RAS oncogene family; RAB15: RAB15, member RAS oncogene family) could distinguish lung cancer patients from healthy individuals [69].

## 3. Actionable Markers for Treatment

### 3.1. Genomics Biomarkers

In the United States, the National Cancer Institute (NCI) and National Human Genome Research Institute launched a genomic cancer program, The Cancer Genome Atlas (TCGA), in 2006. This project characterized 20,000 primary cancer and matched standard samples of 33 major cancer types in the genomic, epigenomic, transcriptomics, clinical, and proteomics information level [77]. Over several years, TCGA data has been available in several databases, such as cBioPortal for Cancer Genomics and The Cancer Proteome Atlas Portal (TCPA) [78,79]. The Catalogue of Somatic Mutations in Cancer (COSMIC) and the International Cancer Genome Consortium (ICGC) provide a comprehensive genomic data visualization, integration, and analysis source [80,81]. Integrative analysis from the Pan-Cancer Analysis of Whole Genomes (PCAWG) Consortium of the International Cancer Genome Consortium (ICGC) and TCGA revealed that cancer genomes contained four to five driver mutations [82]. Recent results from the PCAWG Consortium shed light on different aspects of cancers, such as mutational processes, tumor evolution, and diverse transcriptional consequences [82]. The combination of omics data from various databases such as Gene Expression Omnibus and TCGA is one of the promising approaches to finding potential biomarkers for understanding cancer development and progression [83].

Genetic variations such as single-nucleotide polymorphisms (SNPs) and deletion variants (indels) are associated with genetic susceptibility to cancer [84,85]. Pharmacogenetics studies explore genetic variations and focus on individual genetic variability and its correlation with drug efficacy, toxicity, and overall cancer treatment outcomes [86,87].

Several driver gene alterations have been detected in NSCLC patients, including Epidermal growth factor receptor (*EGFR*), Kirsten rat sarcoma viral oncogene homolog (K-Ras), tumor protein p53 (TP53), MET, B-Raf proto-oncogene, serine/threonine kinase (BRAF), ERBB2, the anaplastic lymphoma kinase (ALK), ROS proto-oncogene 1 receptor tyrosine kinase (ROS), RET, and NTRK rearrangement were detected [88,89,90,91].

Epidermal growth factor receptor (*EGFR*) controls critical signaling pathways, including RAS/MAPK/ERK, PI3K/AKT, and STATS, and is responsible for cellular proliferation, differentiation, and survival. *EGFR* is an oncogenic driver in NSCLC, breast, and glioblastoma. *EGFR* has been linked to types of cancer, including metastatic colorectal cancer (CRC), pancreatic cancer, and head and neck squamous cell carcinoma (HNSCC). Its gene is located on the short arm of chromosome 7 at position 11.2 (7p11.2) [92]. *EGFR* is overexpressed and/or mutated in cancers such as glioblastoma, head and neck, pancreatic, breast, and metastatic colorectal cancer [93,94,95,96,97]. *EGFR* is overexpressed in 40–80% of NSCLC patients [98,99]. Abnormal *EGFR* expression has been associated with mutations in the *EGFR*, with approximately 10–15% in Caucasians with adenocarcinoma and 50% in lung tumors in East Asian populations [100]. Common mutations of *EGFR* are summarized in Figure 2. Several tyrosine kinase inhibitors (TKIs) targeting *EGFR* have been developed, including Erlotinib (Tarceva), Afatinib (Gilotrif), Gefitinib (Iressa), Osimertinib (Tagrisso), and Dacomitinib (Vizimpro), which improved progression-free survival (PFS), time-to-progression, and overall survival compared to standard chemotherapy in NSCLC patients. Gefitinib, Erlotinib, and Afatinib are common options for lung cancer patients with exon 21 Leu858Arg mutation or *EGFR*-exon 19 deletions. Unfortunately, patients with *EGFR*-mutant NSCLC develop disease progression within 10–14 months [101]. Different mechanisms underlie intrinsic or acquired resistance to EGFR-targeted inhibitors. T790M mutation in *EGFR* is responsible for approximately 50% of cases of acquired resistance [102]. Other important resistance mechanisms involve AKT mutations, loss of PTEN, activation of alternative signaling, and BCL2-like 11/BIM deletion polymorphism [103].

The *ALK* gene consists of 30 exons and is located on chromosome 2p23. *ALK* is a tyrosine kinase receptor of the insulin receptor superfamily that plays an essential role in embryonic and neural development and human immune responses. The role of *ALK* in human tumorigenesis was discovered in anaplastic large cell lymphoma (ALCL) in 1994. The presence of EML4-ALK fusion was shown in breast, colorectal, and non-small-cell lung cancers. *KIF5B*, *RET*, *ROS1*, and ALK fusion has been detected in lung cancer. *ALK* could have different fusion partners, including NPM, in non-Hodgkin’s lymphoma, VCL in Renal cell carcinoma, FN1 in gastrointestinal leiomyomas, and TRK-fused gene (TFG) in anaplastic large-cell lymphoma.

The V-RAF murine sarcoma viral oncogene homolog B (*BRAF*) encodes serine/threonine kinase. *BRAF* is a constituent of the MAPK pathway that regulates various cellular processes, including cell growth, proliferation, and survival. The effect of *BRAF* mutations in lung cancer patients was first reported in 2011 [104]. Mutations in BRAF are seen in 2–5 lung adenocarcinomas [105].

*ROS1* (ROS proto-oncogene 1) is located on chromosome 6p22 and encodes 2347 amino acid residues. *ROS1* is the subfamily of receptor tyrosine kinase, which regulates cell proliferation, survival, and migration [106]. FIG (GOPC)-*ROS1* was the first *ROS1* fusion gene reported in the human glioblastoma cell line U118MG in the early 1980s. Genetic rearrangements of *ROS1* have been shown in different malignancies, including gastric adenocarcinoma, colorectal cancer, ovarian cancer, inflammatory myofibroblastic tumor, and angiosarcoma [106,107,108,109,110]. The first *ROS1* fusion gene in NSCLC was identified in 2007. Previous studies have shown that *ROS1* fusions account for 1–2% of all cases of NSCLC [88]. NSCLC patients with *ROS1* fusion have several distinct clinical characteristics: they are typically female, younger (<50 years of age), and never or light smokers [111,112,113].

#### 3.1.1. The PI3K Pathway in NSCLC

The phosphatidylinositol 3-kinase (PI3K)/Akt/mechanistic target of the rapamycin (mTOR) pathway has a pivotal role in regulating signal transduction and biological processes such as cell growth and survival, apoptosis, angiogenesis, tumor invasion, and metastasis. PI3Ks, AKT, and mTOR are the core components of the PI3K–AKT–mTOR signaling pathway, and their hyperactivation is observed in many cancers [114,115]. Therefore, alterations in this pathway lead to initiating and enhancing cancer progression. Consequently, this pathway is considered a target for novel anticancer therapies.

In NSCLC, the PI3K/Akt/mTOR pathway has been heavily implicated in carcinogenesis and disease advancement in NSCLC. Mutations in receptor tyrosine kinases (RTKs) and KRAS have been linked to the activation of the PI3K/Akt/mTOR axis in NSCLC [116,117]. Alterations in each component of this pathway may lead to lung cancer development and progression [118]. Mutations or increases in the number of PIK3CA of *PIK3CA* are frequently found in patients with NSCLC. *PIK3CA* mutations are one of the most common gene changes in human cancers. Mutations or increases in the number of PIK3CA of *PIK3CA* are frequently found in patients with NSCLC.

A large study by Scheffler et al. analyzed 1144 consecutive NSCLC patients’ tumor tissue for *PIK3CA* mutations. This study showed that 3.7% of patients have PIK3CA mutations in exons nine and 20 [118,119]. Yamamoto et al. investigated *PIK3CA* mutations in exons 9 and 20 in 86 NSCLC cell lines and 356 resected NSCLC tumors and revealed that *PIK3CA* mutations in NSCLC cell lines were 4.7% and tumors 1.6%. Increased *PIK3CA* copy number was detected in 9.3% of NSCLC cell lines and 17.1% of NSCLC tumors [120].

Recurrent alterations were observed in *PIK3CA* that were considered to contribute Osimertinib resistance [121]. A systematic review and meta-analysis study was conducted with a total of 13 studies involving 3908 NSCLC patients. The results indicated that PIK3CA mutations are associated with OS, PFS, and cancer-specific survival (CSS) [122]. PIK3CA mutation could be considered an independent prognostic factor for reduced PFS of EGFR-TKIs treatment and worse OS in NSCLC patients [122,123].

Several PI3K/Akt/mTOR signaling pathway inhibitors are currently developing as promising targets for new anticancer drugs in preclinical investigations and clinical trials. Pilaralisib is a highly selective inhibitor of the class I PI3Ks and successfully inhibits tumor growth in vivo. In phase I trials, Pilaralisib was assessed as a tolerable monotherapy in advanced solid tumors patients [109]. Alpelisib is a p110 alpha isoform-specific PI3K inhibitor that has been FDA-approved for HER2-positive advanced or metastatic breast cancer with PIK3CA-mutated. Alpelisib has been experimentally used in solid tumors with a favorable safety profile [109,124].

However, there is limited data on using *PIK3CA* inhibitors such as Copanlisib, Idelalisib, Umbralisib, Duvelisib, and Alpelisib in NSCLC patients. There is a need to elucidate the underlying molecular biology to detect the relevant biomarkers of toxicities and resistance to PI3K therapies. Several NGS-based panels related to cancer were approved by the FDA. Oncomine Dx Target Test detects variations in 23 genes that are biomarkers for selecting NSCLC patients for four targeted therapies Dabrafenib, Crizotinib, Gefitinib, and Pralsetinib [125]. The Praxis Extended RAS Panel detects several mutations in exons 2, 3, and 4 of *KRAS* and *NRAS* to implicate metastatic colorectal cancer treatment decisions [126]. MSK-IMPACT was created by the Memorial Sloan Kettering Cancer Center (MSK) to detect genetic aberrations in the 468-gene in solid tumors [127]. The FoundationOne CDx diagnostic test detects genetic mutations in 324 genes in several FDA-approved therapies for solid tumors [128]. Ongoing clinical trials are summarized in Table 4.

Of note, Crizotinib is an ALK, MET, and ROS1 kinases inhibitor. The phase I study of Crizotinib in 50 patients who were positive for ROS1 rearrangement proved the antitumor activity of this drug in advanced NSCLC [129]. Consequently, in 2016, Crizotinib was approved by the U.S. Food and Drug Administration (FDA) for the treatment of advanced NSCLC patients with *ROS1*-rearranged. In a phase II study of Crizotinib in East Asian patients, 127 East Asian patients were enrolled with *ROS1*-positive. The ORR by IRR was 71.7% (95% CI, 63.0% to 79.3) for NSCLC patients with *ROS1*-positive and had lower rates of brain metastases compared to *ALK* rearrangements (*ROS1* 19.4%, *ALK* 39.1%; *p* = 0.033) [130]. Another study identified 33 *ROS1* rearrangements in 579 patients with stage IV NSCLC. The median PFS time for *ROS1*-positive was 11 months. *ROS1* fusion partners were *CD74*, *SLC34A2*, and *ZCCHC8* [131]. In Chinese patients with *ROS1*-positive advanced NSCLC treated with Crizotinib, the objective response rate (ORR) and the disease control rate (DCR) were 71.4% and 94.3%, respectively. In addition, the median PFS was 11.0 months (95% confidence interval (CI), 7.8–14.2), and median OS was 41.0 months (95% CI, 22.5–59.5) [132].

#### 3.1.2. Current Status of Novel Biomarkers for Response to Immunotherapy

Immune checkpoint inhibitors (ICIs) are considered one of the most important treatments for various advanced malignancies and have become the standard of care in lung cancer [133,134]. Among ICI immunotherapy agents, the antiprogrammed death 1 ligand (PD-L1) (atezolizumab and durvalumab), the cytotoxic T-lymphocyte antigen 4 (anti-CTLA-4) (ipilimumab), and the programmed cell death receptor-1 (PD-1) (nivolumab and pembrolizumab) are currently used in therapy in patients with lung cancer. Treatment outcomes of lung cancer patients have improved considerably with these drugs; however, these drugs can cause immune-related adverse events (irAEs) that threaten the life of patients. Moreover, only a relatively small number of lung cancer patients benefit from immunotherapy, and the rapid tumor progression after the treatment with ICIs defined is as a hyperprogressive disease (HPD). Hence, an urgent need for biomarkers that predict responses and prognosis after treatment with ICIs is warranted.

Previous large clinical trials reported that the higher expression of PD-L1 can predict the response to PD1/PD-L1 inhibitors in different types of cancer, such as non-small-cell lung cancer (NSCLC), urothelial cancer, and melanoma [132,135,136,137]. Immunohistochemical (IHC) detection of PD-L1 is a current predictive biomarker, but it is an imperfect biomarker for responses to ICI therapy.

In the KEYNOTE-001 trial, patients with advanced NSCLC enrolled in a phase I study and received Pembrolizumab. The PD-L1 expression in the tumor samples using the IHC analysis and results declared that at least 50% of the PD-L1 expression in the tumor cells correlated with improved efficacy of Pembrolizumab in patients [138]. In the open-label, phase II randomized controlled POPLAR trial, improvement of NSCLC patients treated with Atezolizumab was correlated with PD-L1 expression [139]. Data from the KEYNOTE-024 study reported that advanced NSCLC patients with PD-L1 expression on at least 50% of tumor cells treated with Pembrolizumab were associated with significantly longer PFS and OS [136]. However, there are some challenging obstacles to consistency in PD-L1 testing, including sampling methods, different techniques used to assess PDL1 expression, dynamic changes in PD-L1 expression, and spatial and temporal heterogeneity [122]. In addition, the single biomarker may not be indicative of patient selection. Therefore, several predictive and prognostic biomarkers have been explored during the past decade to guide patient choice.

#### 3.1.3. Tumor Mutation Burden (TMB) and Circulating Tumor DNA (ctDNA)

The tumor mutation burden (TMB) refers to the total number of somatic mutations in tumor cells. The first association between a high TMB and treatment response was reported in patients with melanoma treated with cytotoxic T-lymphocyte-associated protein 4 (CTLA-4) [140]. Recently, various clinical trials showed that TMB is an important biomarker for predicting the efficacy of immunotherapy across diverse tumor types, including lung, urothelial, and breast cancers [141,142]. New studies have shown that blood-based TMB (bTMB) from circulating tumor DNA (ctDNA) has predictive power for the clinical benefit of immunotherapy in non-NSCLC patients receiving immunotherapy [142,143]. The role of bTMB profiling and sequencing of small amounts of cell-free DNA (cfDNA) in NSCLC patients treated with ICIs has been explored in several studies. Gandara et al. tested a novel assay to measure bTMB using samples collected prospectively from the POPLAR (NCT01903993) and the OAK (NCT02008227) clinical trials. Their results indicated that high bTMB in patients receiving Atezolizumab monotherapy in NSCLC was associated with longer PFS [143]. Wang et al. explored a 150-cancer genes panel (CGP) named NCC-GP150 in blood and tissue samples from non-small-cell lung cancer treated with anti-PD-1 and anti-PD-L1 therapy. Data from this analysis demonstrated that bTMB was a potential biomarker for patients with advanced NSCLC receiving ICIs [144]. Iijima et al. studied early response in 14 NSCLC patients treated with Nivolumab. The high level of tumor burden in circulating tumor DNA (ctDNA) could predict a durable response to Nivolumab [145]. Raja et al. studied variant allele frequencies (VAF) of somatic mutations in 73 genes ctDNA of NSCLC and urothelial cancer patients. Patients with the reduction in ctDNA VAF had a more significant decrease in tumor volume, with longer PFS and OS. [146]. In the study of Guibert et al., targeted sequencing of plasma ctDNA was analyzed in 97 progressive NSCLC patients. This study revealed that if a ctDNA allele fraction (A.F.) decreases, the PFS of patients increases. B-F1RST is the first prospective trial that evaluated bTMB in advanced NSCLC treated with Atezolizumab as a first-line (*1L*) monotherapy. The final result of this trial confirmed that bTMB could be considered a predictive biomarker for PFS and OS in NSCLC patients [146,147].

Atezolizumab monotherapy resulted in longer overall survival in NSCLC patients with high PD-L1 expression [148]. In the phase II B-F1RST (NCT02848651) trial, Kim et al. evaluated bTMB in locally advanced or metastatic stage IIIB–IVB NSCLC patients treated with first-line Atezolizumab monotherapy. In patients with bTMB ≥ 16, OS were associated with longer than patients with bTMB < 16 [142].

In meta-analyses of Meng et al., predictive value of TMB in NSCLC patients treated with ICIs were analyzed in six randomized controlled trials (3662 patients) and 31 datasets (3437 patients). Longer PFS, OS, and higher ORR were observed in high-TMB patients. Furthermore, immunotherapy was associated with improved PFS, OS, and ORR in patients with high-TMB patients [149].

In the systematic review and meta-analysis of Kim et al., they explored the data of 5712 patients from 26 studies. The result of this study showed that, in patients who were treated with ICIs, the high-TMB groups showed better OS and PFS compared to the low-TMB groups. Moreover, patients with high TMB benefit more from ICI treatment than chemotherapy alone. In NSCLC patients, higher TMB was significantly associated with better PFS [150].

In the study of Thompson et al., ctDNA NGS was identified 50 drivers and 12 resistance mutations in advanced NSCLC. Detection of targetable driver and resistance mutations in ctDNA led to the identification of the resistance mechanisms in patients and to finding additional therapeutic options [151].

### 3.2. Proteomics

Protein markers can monitor treatment and/or predict drug resistance in NSCLC patients. Xu et al. utilized a SILAC-based quantitative proteomic approach to elucidate the Paclitaxel (PTX) resistance. They found that PDCD4 mediated PTX sensitivity in cancer cell lines, and during adjuvant therapy of lung cancer patients with PTX, the overall survival of patients correlated with high levels of PDCD4 [152]. Programmed cell death 4 (PDCD4) is downregulated in different cancer types and is considered a tumor suppressor. It has been proved that Pdcd4 has various functions inhibiting cell growth, suppressing tumor promotion, metastasis, and inducing apoptosis. For instance, the loss of PDCD4 expression is correlated with tumor progression and prognosis and survival in lung, breast, ovarian, and colorectal cancers. In lung cancer cells, Vihreva et al. demonstrated that the transcription of *PDCD4* is negatively regulated by mTOR signaling.

Sandfeld-Paulsen et al. studied exosomal membrane proteins by using the extracellular vesicle array in 276 NSCLC patients. They found increasing concentration levels of NY-ESO-1, EGFR, PLAP, EpCam, and Alix have been associated with inferior OS [153]. In the same year, they published another study isolating exosomes from the blood of 581 patients (431 with lung cancer and 150 controls). They found that markers CD151, CD171, and tetraspanin eight could significantly determine NSCLC patients versus healthy controls [154]. In the PROSE study, Gregorc et al. demonstrated that a serum protein test could predict survival in non-small-cell lung cancer treated with Erlotinib and single-agent chemotherapy [155]. In the study by Salmon et al., L.C. patients were treated with Erlotinib in combination with Bevacizumab, and matrix-assisted laser desorption ionization time-of-flight mass spectrometry (MALDI-*TOF* MS) could classify patients who benefit from treatment with Erlotinib [155,156]. VeriStrat is a plasma- or serum-based test utilizing MALDI MS methods to monitor EGFR TKIs treatment of various tumor types, including colorectal and head and neck cancer [157]. Multiple studies demonstrated that the blood-based proteomic VeriStrat test could help predict EGFR TKI treatment response. In lung cancer, Taguchi et al. analyzed serum proteins by MALDI MS in NSCLC patients treated with EGFR TKIs. Proteomic evaluation of this study classified patients for good or poor outcomes after treatment with Gefitinib or Erlotinib [157,158]. Plasma samples of 111 NSCLC patients were analyzed by MALDI-*TOF* MS and ranked by the VeriStrat test [159]. Patients in the VeriStrat “good” classification had longer PFS and OS. Carbone et al. analyzed a plasma sample of 441 NSCLC patients in the NCIC Clinical Trials Group BR.21 trial. According to the results, the VeriStrat test predicts the objective response to Erlotinib and is a prognostic for both OS and PFS [160]. Fidler et al. studies the serum samples of NSCLC patients who received chemotherapy or Erlotinib. They measured 102 biomarkers by Luminex immunoassays and classified patients with the VeriStrat test. This study showed 27 and 16 biomarkers associated with OS and PFS, respectively [161]. In the study of Grossi et al., 481 NSCLC patients were treated with first-line platinum-based chemotherapy. In the group receiving VeriStrat, the PFS and OS were longer [161,162].

## 4. AI Machine Learning-Driven Discovery of Biomarkers for NSCLC

Robust clinical evaluation requires innovations, including technology platforms and artificial intelligence (AI). AI mimics the problem-solving and decision-making capabilities of the human mind [163]. Machine learning (ML) and its subfield deep learning are the main subsets of AI that can help biologists analyze vast amounts of data, leverage patterns in the data, and transform big data into actionable biomarkers [164]. AI revolutionized various areas in medicine, such as ophthalmology, radiology, and oncology [165,166,167]. Several medical algorithms detect CT-based lesions and are approved by the FDA, such as Arterys Oncology DL, Arterys MICA, and QuantX™, that improve radiologist performance [168,169]. AI affects different aspects of lung cancer, such as medical imaging for screening, early detection, characterizing lung cancers, treatment selection, and monitoring treatment response [170,171]. However, further improvement is required in big data interpretation and management, model fairness and interpretability, and generalizability.

Artificial intelligence has many applications in lung cancer [172]. Zhang et al. found that a combination of five urinary biomarkers discriminated lung cancer patients from healthy ones and differentiated lung cancer from other cancers [69]. Analysis of microbial reads of TCGA of 33 cancer types with the stochastic gradient-boosting machine (GBM) learning models successfully discriminated against cancer-free individuals, patients with cancer, and patients with multiple types of cancer [173]. In the study of Tirzïte et al., lung cancer patients’ exhaled breath samples were explored with logistic regression analysis (LRA). This model can discriminate lung cancer patients from noncancer patients [174]. It is trained on a deep convolutional neural network (inception v3) that could predict most STK11, EGFR, FAT1, SETBP1, KRAS, and TP53 from pathology images [175]. The PD-L1 expression level is a crucial biomarker for identifying responders and nonresponders to NSCLC patients treated with anti-PD-1/PD-L1 treatments. However, the accurate estimation of PD-L1 expression is challenging. Wang et al. proposed a deep learning model that predicted mutated EGFR from the wild EGFR patients and distinguished PD-L1-positive from PD-L1-negative patients through C.T. images [176]. Choi et al. developed an AI-powered tumor proportion score (TPS) analyzer to evaluate whole-slide images of 802 NSCLC. Using this model, they could detect PD-L1 expression in tumors and calculate TPS. Moreover, AI-assisted TPS reading predicted OS and PFS upon ICI treatment [177]. Cheng et al. developed deep learning (DL)-based AI model to analyze the expression of PD-L1 in 1288 lung cancer patients. They used three different AI models (M1, M2, and M3) assessed in both PDL1 (22C3) and PD-L1 (SP263) assays. Their models improved the evaluation of PD-L1 expression, and diagnostic results were consistent with the pathologist’s [177,178]. In the study of Wu et al., TPS of PD-L1 expression was also assessed in whole slide images (WSIs) of the 22c3 assay by the DL model. Their results also confirmed that AI-assisted diagnosis tests improve diagnostic repeatability and are a promising tool for improving the efficiency of clinical pathologists [179].

## 5. Conclusions and Future Directions

Cancer is a heterogeneous disease caused mainly by genomic aberrations. These aberrations may alter proteome, transcriptome, and metabolome pathways, which in turn become valuable biomarkers. Genomics and proteomics heterogeneity is the primary barrier that may be responsible for developing drug resistance. Using new techniques, such as artificial intelligence or single cell RNA sequencing, to solve old problems could solve old problems such as discovering biomarkers for diagnostic, prognostic, and predictive biomarkers to help guide clinicians to choose their medicine more precisely. These data layers integrate advanced computational methods to overview tumorigenesis comprehensively. Rethinking with new methods the resistance to checkpoint inhibitors (e.g., PD-1, PD-L1 or CTLA-4) or targeted therapies (e.g., PI3K/mROT pathways inhibitors) could help in selecting only patients who would benefit from these therapies in the future.

Solid high-throughput clinical outcomes are pivotal for clinical biomarkers discovery. Therefore, international collaboration with a large sample size of a wide variety of cancers and sharing of treatment data is imperative to develop more accurate biomarkers predicting the therapy response of lung cancer patients. Early detection and a proper treatment regimen prediction can ultimately aid clinicians in improving those patients’ survival rates.

## Figures and Tables

**Figure 1 cancers-14-05144-f001:**
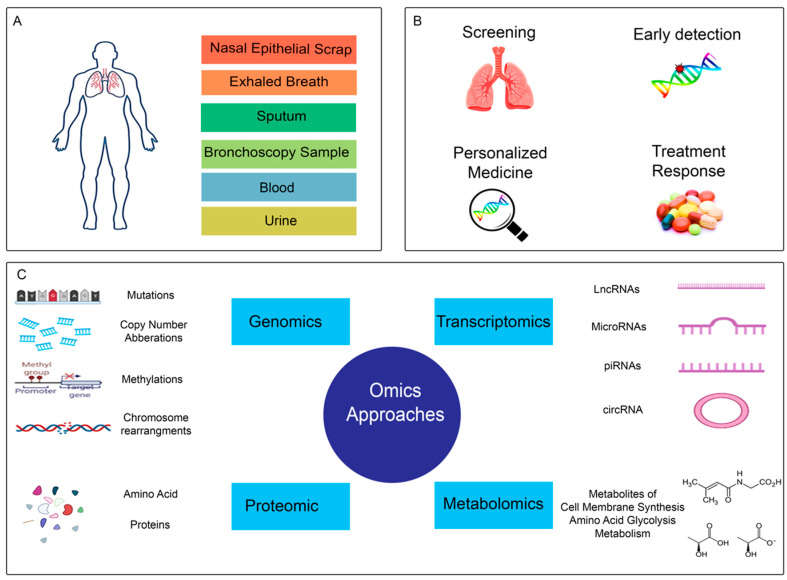
Schematic overview of the NSCLC biomarker discovery. (**A**) Different samples. Nasal epithelial scraping, exhaled breath, sputum, bronchoscopy, blood, and urine are used for NSCLC biomarker discovery. (**B**) NSCLC biomarker applications. The biomarkers can be helpful for screening, early detection, personalized medicine, and monitoring treatment response. (**C**) Different approaches for biomarker discovery. Biomarkers can be discovered through genomic, transcriptomic, proteomic, and metabolomics changes in NSCLC.

**Figure 2 cancers-14-05144-f002:**
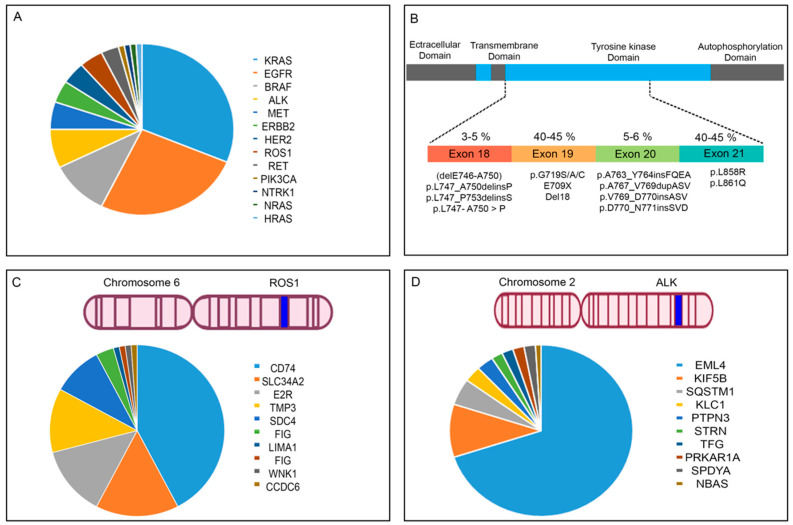
Genetic alteration in NSCLC. (**A**) Frequency of driver genes in NSCLC. (**B**) Frequency of driver mutations in EGFR. Fusion partners of ROS1 (**C**) and ALK (**D**) in NSCLC.

**Table 1 cancers-14-05144-t001:** Methylation-based biomarkers for early detection of NSCLC.

Biomarker (S)	Method	Specimen	Population	Sensitivity and Specificity (%)	Reference
*MGMT*, *p16*, *RASSF1A*, *DAPK*, and *RAR-β*	Meta-analysis	Blood	37 case-control studies	NA	[36]
*APC*, *CDH13*, *KLK10*, *DLEC1*, *RASSF1A*, *EFEMP1*, *SFRP1*, *RARβ* and *p16INK4A*	MSP	Tissues	78 paired NSCLC specimens and adjacent normal tissues 110 stage I/II NSCLC and 50 plasmas cancer-free	83.64 and 74.0	[37]
*RARB2*, *RASSF1A*	Quantitative methylation-specific PCR	Cell-Free DNA circulating in the blood (cirDNA)	32 healthy donors and 60 patients with lung cancer	87 and 75	[38]
*SHOX2*	Quantitative real-time polymerase chain reaction	Plasma	371 samples from patients with lung cancer and controls	60 and 90	[39]
*DCLK1*	qMSP-PCR	Plasma	65 patients with lung cancer and 95 healthy donors	NA	[39,40]
*SEPT9*	Real-time PCR with the use of specific *SEPT9* promoter methylation probe	Plasma	70 lung cancer patients and 100 healthy individuals	44.3 and 92.3	[41]
*CDO1*, *BCAT1*, *TRIM58*, *ZNF177*	Pyrosequencing	Paraffin- embedded tissues Bronchial aspirates and bronchoalveolar lavages	237 stage I NSCLC and 25 nontumoral matched lung tissues	NA	[42]
*TMEFF2*	Methylation-specific PCR	Serum	316 NSCLC, 50 NC	9.2 and 100	[43]

**Table 2 cancers-14-05144-t002:** MicroRNA biomarkers for early detection of NSCLC.

Biomarkers	Specimen	Population	Result	Sensitivity and Specificity (%)	Reference
13 miRNA	Plasma	939 participants, including 69 patients with lung cancer and 870 healthy control subjects	Screening	87 and 81	[53]
miR-31 and miR-210	Sputum	35 patients with lung cancer and 40 healthy control subjects	Screening	65.71 and 85.00	[53,54]
miR-125a-5p, miR-25, and miR-126	Serum	24 early stage lung cancer patients and 24 healthy control subjects	Early Detection	87.5 and 87.5	[49]
miR-21, miR-143, miR- 155, miR-210, miR-372	Sputum	24 NSCLC cases and 6 negative controls	Early Detection	83.3 and 100	[55]
miR-141	Plasma	NSCLC patients (n: 72) and N.C. (n: 50)	Early Detection	82.7 and 98	[56]
miRNA (miR)-486 and miR-150	Peripheral Blood		Early Diagnosis and Recurrence	90.9 and 81.8 for miR-486 and 81.8 for miR-150	[57]
miRs-126, 145, 210, and 205-5p	Plasma	64 individuals comprising 34 lung cancer patients and 30 healthy control smokers	Early Detection	91.5 and 96.2	[58]
I-miR-1254 and hsamiR-574-5p	Serum	22 individuals (11 healthy control subjects and 11 patients with early stage NSCLC).	Early Detection	82 and 77	[58,59]

**Table 3 cancers-14-05144-t003:** Protein biomarkers for early detection of NSCLC.

Biomarker	Method	Specimen	Population	Sensitivity and Specificity (%)	Reference
FTL, FGB, RAB33B, RAB15	LC-MS/MS	Urine	Lung cancers from healthy control subjects	90 and 90	[69]
ERO1L, NARS, PABPC4, RCC1, RPS25, TARS	(iTRAQ) labeling combined with 2D-LCMS/M.S.	Tumor and Lung Tissues	ADC tumors without L.N. metastasis and adjacent normal tissues	NA	[70]
44 proteins showed a fold-change > 3.75	(L.C.–MS/MS)	Bronchoalveolar Lavage Fluid (BALF)	Adenocarcinoma vs. healthy control subjects	NA	[71]
133 biomarkers	LC-MS	Bronchoalveolar Lavage (BAL)	Lung cancer versus nonlung cancer	NA	[71,72]
GlcNAcylated AACT	iTRAQ labeling and LC-MS/MS.	Serum	NSCLC patients, benign lung diseases subjects, and healthy individuals	90.8 and 76.9	[73]
α2 macroglobulin, αmicroglobulin/bikunin, and SERPINA1	MRM	Serum	NSCLC lung adenocarcinoma cancer and healthy control subjects	NA	[74]
Elongation factor 1- alpha 2, proteasome subunit alpha type, and spermatogenesis-associated protein	LC-MS/MS	Serum	Lung cancer and healthy control subjects	NA	[75]
ALOX5, ALOX5AP, SLC2A3, CEACAM6, ITGAX, CRABP2, LAD1	LC-MS, PMR-MS, and immunohistochemistry	Tissues and Normal Bronchial Biopsies	Adenocarcinoma samples and benign nodules	NA	[76]

LC-MS/MS: liquid chromatography–tandem mass spectrometry, PMR-MS: parallel reaction monitoring mass spectrometry, iTRAQ: isobaric tags for relative and absolute quantification MRM: multiple reaction monitoring, NA: not applicable.

**Table 4 cancers-14-05144-t004:** Ongoing clinical trials are investigating treatments targeting the AKT–PI3K–mTOR pathway in NSCLC.

NCT Number	Clinical Phase	Types of Patients	Purpose	Primary End Points	Intervention/s
NCT04467801	II	60 metastatic/advanced NSCLC	Treatment	Progression-free survival	Ipatasertib
NCT04184921	NA	350 advanced lung cancer patients	NA	Progression-free survival	Osimertinib
NCT03543683	NA	330 metastatic NSCLC	NA	1-year median progression-free survival	Osimertinib
NCT03532698	NA	100 stage IIIB and IV NSCLC	NA	Objective response rate (ORR)	Osimertinib
NCT03845270	II	46 stage III or IV NSCLC	Treatment	Overall response	Pertuzumab + Trastuzumab + Docetaxel
NCT01306045	II	471 advanced NSCLC, SCLC, and thymic malignancies	Treatment	Estimate the response rate and feasibility of the use of tumor molecular profiling and targeted therapies in the treatment of NSCLC, SCLC, and thymic malignancies	AZD6244 MK-2206 Lapatinib Erlotinib Sunitinib
NCT02664935	II	423 NSCLC stage III or stage IV	Treatment	Objective response (OR), progression-free survival time (PFS), and durable clinical benefit (DCB)	AZD4547 Vistusertib Palbociclib Crizotinib Selumetinib Docetaxel AZD5363 Osimertinib Durvalumab Sitravatinib AZD6738
NCT02117167	II	999 metastatic relapse or stage IV	Treatment	Progression-free survival	AZD2014 AZD4547 AZD5363 AZD8931 Selumetinib Vandetanib Pemetrexed Durvalumab Savolitinib Olaparib
NCT04591431	II	384 recurrent/metastatic breast, gastrointestinal cancer, non-small-cell lung cancer, or others	Treatment	Overall response rate (ORR)	Erlotinib Trastuzumab Trastuzumab emtansine Pertuzumab Lapatinib Everolimus Vemurafenib Cobimetinib Alectinib Brigatinib Palbociclib Ponatinib Vismogedib Itacitinib Ipatasertib Entrectinib Atezolizumab Nivolumab Ipilimumab Pemigatinib
NCT04467801	II	60 metastatic/advanced NSCLC	Treatment	Progression Free Survival	Ipatasertib
NCT04184921	NA	350 advanced lung cancer	NA	Progression-free survival	Osimertinib
NCT03543683	NA	330 metastatic NSCLC	NA	1-year median progression-free survival (PFS)	Osimertinib
NCT03532698	NA	100 metastatic NSCLC	NA	Objective response rate (ORR)	Osimertinib
NCT03845270	II	46 stage III and metastatic	Treatment	Overall response	Pertuzumab + Trastuzumab + Docetaxel
NCT01306045	II				AZD6244 MK-2206 Lapatinib Erlotinib Sunitinib
NCT02664935	II				AZD4547 Vistusertib Palbociclib Crizotinib Selumetinib Docetaxel AZD5363 Osimertinib Durvalumab Sitravatinib AZD6738
NCT02117167	II				AZD2014 AZD4547 AZD5363 AZD8931 Selumetinib Vandetanib Pemetrexed Durvalumab Savolitinib Olaparib
NCT04591431	II				Erlotinib Trastuzumab Trastuzumab emtansine Pertuzumab Lapatinib Everolimus Vemurafenib Cobimetinib Alectinib Brigatinib Palbociclib Ponatinib Vismogedib Itacitinib Ipatasertib Entrectinib Atezolizumab Nivolumab Ipilimumab Pemigatinib
NCT01737502	I and II	47 lung cancer (squamous, Ras-mutated adenocarcinoma, or small-cell lung cancer)	Treatment	Maximum tolerated dose of Auranofin, number and severity of all adverse events, and progression-free survival	Auranofin Sirolimus
NCT05445791	III				Metformin Hydrochloride
NCT02664935	II				AZD4547 Vistusertib Palbociclib Crizotinib Selumetinib Docetaxel AZD5363 Osimertinib Durvalumab Sitravatinib AZD6738
NCT02117167	II				AZD2014 AZD4547 AZD5363 AZD8931 Selumetinib Vandetanib Pemetrexed Durvalumab Savolitinib Olaparib
NCT04591431	II				Erlotinib Trastuzumab Pertuzumab Lapatinib Everolimus Vemurafenib Cobimetinib Alectinib Brigatinib Palbociclib Ponatinib Vismogedib Itacitinib Ipatasertib Entrectinib Atezolizumab Nivolumab Ipilimumab Pemigatinib
NCT05144698	II	22 advanced metastatic, recurrent, and unresectable solid tumors	Treatment	Safety of RAPA-201 Cell Therapy	RAPA-201 Rapamycin-Resistant T Cells Chemotherapy Prior to RAPA-201 Therapy

AZD2014; Novel mTOR inhibitor; AZD4547; FGFR inhibitor; AZD5363; Akt inhibitor; AZD8931; Novel EGFR/HER2/HER3 signaling inhibitor; NA; Not available.

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
