# Peer review of "The Role of Genomics and Proteomics in Lung Cancer Early Detection and Treatment†"

_cancers, 2022, doi:10.3390/cancers14205144_

Round 1

Reviewer 1 Report

In this Review, Abbasian et al. summarize the genomic and proteomic biomarkers for early detection and treatment of NSCLC. By review of about 166 references, the authors have summarized many aspects of NSCLC biomarkers:

Early detection: Genomic and Epigenomes (Methylation, miRNAs, Proteomic Early Dection

Actionable Markers for Treatment: Genomic biomarkers (EGFR KRAS, MET, BRAF, ERBB2, ALK, ROS1, and et al.), Proteomic biomarker (PDCD4, ESO1, PLAP, CD151 and et al.)

New technology for biomarker dection: AI machine learning-driven discovery of biomarkers

The figures and tables in this manuscript are excellent and easy to understand. Their work is competent for review of NSCLC biomarker and is enough for publication in our journal.

Author Response

Re: Manuscript no # cancers-1932071 Entitled "The Role of Genomics and Proteomics in Lung Cancer Early Detection and Treatment"

Dear Editors in Chief,

We would like to acknowledge reviewers for their comments and suggestions which much helped to improve the manuscript. We have amended the manuscript in line with the reviewers’ suggestions and the revised version of our manuscript is now being re-submitted. Edited parts of the revised manuscript version have been highlighted with track change. Please find here below our answers to all referees comments, presented in the same order as in the review. Also, the manuscript have polished terms of English proficiency with the help of Dr. Navid Sobhani, who has studied in the United Kingdom for 5 years for his first degrees and his English is very fluent. We hope that the manuscript is now suitable for publication in "Cancers".

We are looking forward to hearing from you in due course the final decision on our article.

Yours sincerely,

Raheleh Roudi

Department of Radiology, Molecular Imaging Program at Stanford, Stanford University, Stanford, CA 94305, USA

E-Mail: roudi@stanford.edu

Reviewer #1:

  1. On page 2, the authors start off about molecular profiling. Building up to the aim of the review, there is no mention, whatsoever, about proteomics. This is important as the title primes the reader to look for a brief background on the role of proteomics in lung cancer. Please add brief part about proteomic before moving to the review aim. It will help reader understand why the authors want to specifically talk about proteomics.

Reply: Thanks for raising this point. This part added in the manuscript, as follows;

Proteins are functional products from the genome. The dynamic range of protein concentration differs from cell to cell and from time to time. Therefore, Proteomics has dynamic and complex entities. A quantitative understanding of proteomics provides the exact knowledge about protein networks of human cells and may improve our cancer detection and treatment tools [15].

Application of proteomics resulted in subtype-specific, stage-specific, and metastasis-specific tumor biomarkers in lung cancer. Numerous studies have demonstrated that a significant number of proteins, including ENO1, selenium-binding protein 1 (SELENBP1), carbonic anhydrase (CA), heat shock 20KD-like protein, and transgelin (SM22-alpha), are associated with poor prognosis [16,17].

  1. On page 2, in the sentence “Furthermore, we discuss recent advances of biomarkers that could be useful in future to predict checkpoint immunotherapies” please add the word “response to” before checkpoint inhibitors.

Reply: Done.

  1. On page 2, the authors state “Five years of survival in NSCLC patients are approximately 63% and 7% for localized and distant tumors, respectively. Early detection of lung cancer will dramatically reduce the mortality rate of patients. Many genomic and proteomic biomarkers such as DNA methylation, microRNA (miRNA), antitumor antibodies, and plasma proteins have been investigated for lung cancer early detection. Liquid biopsy comprises various sources, including sputum, blood, exhaled breath, sputum, bronchial aspirate, and bronchoalveolar lavages for early detection of lung cancer” Please insert reference for these statements.

Reply: Done.

  1. On page 3, under the subheading “Methylation”, the author is talking about the several methylation genes. It is suggested to add briefly about the mechanism of action of these markers and their correlation with early detection or response to treatment in lung cancer.

Reply: Thanks for raising this point. This part added in the manuscript, as follows;

DNA methylation (hypermethylation or hypomethylation) is an epigenetic modification that plays a pivotal role in silencing gene transcription and is crucial in many biological processes, including embryonic development, X-chromosome inactivation, genomic imprinting, chromatin structure, normal growth, and cellular proliferation. Abnormal DNA methylation has been observed in tumor initiation and proliferation of all forms of cancers [22]. The epigenetic pathway is potentially reversible and is involved in tumorigenesis by three main mechanisms DNA hypermethylation, global hypomethylation of the genome and histone modifications.

DNA methylation can be assessed by different techniques, such as digital PCR (dPCR), methylation-specific PCR (MSP), methyl-CpG-binding domain (MBD), realtime quantitative MSP, quantitative methylation-specific PCR (qMSP), multiplex nested methylation-specific PCR, and whole-genome bisulfite sequencing (WGBS) [23]. Aberrant DNA methylation is catalyzed by DNA methyltransferases (DNMTs) through the promoter methylation of tumor suppressor genes. It is associated with many types of cancers, including NSCLC, autoimmune diseases, diabetes, and multiple sclerosis [24–26]. Hypermethylation of DNA can silence tumor suppressor genes, and hypomethylation of DNA can lead to transcriptional activation proto-oncogenes. Methylation patterns have been associated with different cancer aspects, such as disease stage, survival, and response to treatment. DNA Methylation could predict response to platinum-based chemotherapy in lung cancer patients [27,28].

  1. In table 1, alignment is needed, It is suggested that in Table 1, the authors should indicate whether these markers are related to response to treatment or screening. In its present state, the Table specifies randomly about this under the heading of results (which is not appropriate). I would also suggest to add add sensitivity and specificity in Table 1 , if possible as it shows the impact of these markers for lung cancer

 Replay: We appreciate the reviewer’s careful advice. This part modified in the table 1.

  1. Please note that in the text miRNA should not be denoted as MiRNAs

Reply: Done.

  1. On page 5, the authors state “MiRNAs play a pivotal role in biological processes such as cell proliferation, differentiation, and death. Please rephrase the word death with survival.

Reply: Done.

  1. Table 2 is a very busy table. The wuthors keep repeating the word specificity and sensitivity. It is adviced that Specificty and sensitivity should be written in the heading with the % in bracket so that only numbers can be depicted within the table. Also, the table alignment needs attention.

Reply: Done.

  1. The heading of methodology in Tale 2 is not important as it is well known that miRNAS are detected via this methodology. Kindly remove.

Reply: Done.

  1. In Table 2, the authors are using various terms such as disease free, cancer free, negative controls, healthy donor. Please make sure to use consistent terminology such as diseases free or healthy controls to make the table less confusing.

Reply: Done.

  1. In Table 3, please add sensitivity and specificity, if available

Reply: Done.

  1. In Table 4, please remove the status as it does not give the reader any information to know the recruiting/ non-recruiting status in clinical trials. The main information needed by reader about the type of patients enrolled and the end point (whether the trials was looking for screening, prognosis or prediction of response etc.). Kindly add this information in the table.

Replay: We appreciate the reviewer’s careful advice. This part modified in the table 4.

  1. Some of the interventions written in Table 4 are given as “AZD2014 AZD4547 AZD5363 AZD8931”. These seem to be anonymous drug ID. However, if the authors can find the name of the molecule or target molecule, they should insert here, as this information is otherwise useless for the reader.

Reply: Thanks for raising this point. This part added in the footprint of the table 4.

  1. In table 4, Some of the clinical phases are written as “NA”. please write here if these are pre-clinical or in vitro studies. Or indicate in the footer what is meant by NA in the table.

Reply: Done.

  1. On page 7, Actionable Markers for Treatment are discussed. This is well known information that has been reported in many published literatures. It is suggested to discuss any newly discovered markers that can be used as Actionable Markers for Treatment otherwise, this information is unnecessary. In lieu of this, Fig 2 would be useful only of newly discovered markers are discussed.

Reply: Thanks for raising this point. This part was modified in the manuscript.

  1. On page 15, the authors state that “PDCD4 mediated PTX sensitivity in cancer cell lines and during adjuvant therapy of lung cancer patients with PTX, overall survival of patients correlated with high levels of PDCD4”. The readers need a brief background on PDCD4 to understand its impact in lung cancer disease dynamics. Kindly add a little about PDCD4.

Reply: Thanks for raising this point. This part was added in the manuscript, as follows;

Programmed cell death 4 (PDCD4) is down-regulated in different cancer types and is considered a tumor suppressor. It has been proved that Pdcd4 has various functions inhibiting cell growth, suppressing tumor promotion, metastasis, and inducing apoptosis. For instance, the loss of PDCD4 expression is correlated with tumor progression and prognosis and survival in lung, breast, ovarian and colorectal cancers. In lung cancer cells, Vihreva et al. demonstrated that the transcription of PDCD4 is negatively regulated by mTOR signaling.

  1. On page 14, authors talk about bTMB and ctDNA as novel markers. These should be subdivided in headings and more information on these markers should be given as these are new markers and have an impact on reader knowledge.

Reply: Thanks for raising this point. This part was added in the manuscript, as follows;

Atezolizumab monotherapy resulted in longer overall survival in NSCLC patients with high PD-L1 expression[148]. In the phase 2 B-F1RST (NCT02848651) trial, Kim et al. evaluated bTMB in locally advanced or metastatic stage IIIB–IVB NSCLC patients treated with first-line atezolizumab monotherapy. In patients with bTMB ≥ 16, OS were associated with longer than patients with bTMB < 16 [142].

In meta-analyses of Meng et al, predictive value of TMB in NSCLC patients treated with ICIs were analyzed in 6 randomized controlled trials (3662 patients) and 31 datasets (3437 patients) and 6 randomized controlled trials (3662 patients). Longer PFS, OS and higher ORR were observed in high TMB patients. Furthermore, immunotherapy was associated with improved PFS, OS and ORR in patients with high TMB patients [149].

In a systematic review and meta-Analysis of Kim et al, they explored data of 5712 patients from 26 studies. The result of this study showed that in patients who were treated with ICIs, high TMB groups showed better OS and PFS compared to low TMB groups. Moreover, patients with high TMB benefit more from ICI treatment than chemotherapy alone. In NSCLC patients, higher TMB was significantly associated with better PFS  [150].

In the study of Thompson et al. ctDNA NGS was identified 50 drivers and 12 resistance mutations in advanced NSCLC. Detection of targetable driver and resistance mutations in ctDNA, lead to to identify resistance mechanisms in patients and find additional therapeutic options [151].

Reviewer 2 Report

The review article by Hadi Abbasian et al., aims to summarize genomics and proteomics biomarkers for early detection and treatment of NSCLC. The review article overall is well written. However, there is some information that is well known already in literature. The readers would benefit greatly if updated literature on novel or upcoming markers in the pipeline be discussed in detail. Also, there are some suggestions regarding the article (given below):

On page 2, the authors start off about molecular profiling. Building up to the aim of the review, there is no mention, whatsoever, about proteomics. This is important as the title primes the reader to look for a brief background on the role of proteomics in lung cancer. Please add brief part about proteomic before moving to the review aim. It will help reader understand why the authors want to specifically talk about proteomics.

On page 2, in the sentence “Furthermore, we discuss recent advances of biomarkers that could be useful in future to predict checkpoint immunotherapies” please add the word “response to” before checkpoint inhibitors.

On page 2, the authors state “Five years of survival in NSCLC patients are approximately 63% and 7% for localized and distant tumors, respectively. Early detection of lung cancer will dramatically reduce the mortality rate of patients. Many genomic and proteomic biomarkers such as DNA methylation, microRNA (miRNA), antitumor antibodies, and plasma proteins have been investigated for lung cancer early detection. Liquid biopsy comprises various sources, including sputum, blood, exhaled breath, sputum, bronchial aspirate, and bronchoalveolar lavages for early detection of lung cancer” Please insert reference for these statements.

On page 3, under the subheading “Methylation”, the author is talking about the several methylation genes. It is suggested to add briefly about the mechanism of action of these markers and their correlation with early detection or response to treatment in lung cancer.

In table 1, alignment is needed, It is suggested that in Table 1, the authors should indicate whether these markers are related to response to treatment or screening. In its present state, the Table specifies randomly about this under the heading of results (which is not appropriate). I would also suggest to add add sensitivity and specificity in Table 1 , if possible as it shows the impact of these markers for lung cancer

Please note that in the text miRNA should not be denoted as MiRNAs

On page 5, the authors state “MiRNAs play a pivotal role in biological processes such as cell proliferation, differentiation, and death. Please rephrase the word death with survival.

Table 2 is a very busy table. The wuthors keep repeating the word specificity and sensitivity. It is adviced that Specificty and sensitivity should be written in the heading with the % in bracket so that only numbers can be depicted within the table. Also, the table alignment needs attention.

The heading of methodology in Tale 2 is not important as it is well known that miRNAS are detected via this methodology. Kindly remove.

In Table 2, the authors are using various terms such as disease free, cancer free, negative controls, healthy donor. Please make sure to use consistent terminology such as diseases free or healthy controls to make the table less confusing.

In Table 3, please add sensitivity and specificity, if available

In Table 4, please remove the status as it does not give the reader any information to know the recruiting/ non-recruiting status in clinical trials. The main information needed by reader about the type of patients enrolled and the end point (whether the trials was looking for screening, prognosis or prediction of response etc.). Kindly add this information in the table.

Some of the interventions written in Table 4 are given as “AZD2014 AZD4547 AZD5363 AZD8931”. These seem to be anonymous drug ID. However, if the authors can find the name of the molecule or target molecule, they should insert here, as this information is otherwise useless for the reader.

In table 4, Some of the clinical phases are written as “NA”. please write here if these are pre-clinical or in vitro studies. Or indicate in the footer what is meant by NA in the table.

On page 7, Actionable Markers for Treatment are discussed. This is well known information that has been reported in many published literatures. It is suggested to discuss any newly discovered markers that can be used as Actionable Markers for Treatment otherwise, this information is unnecessary. In lieu of this, Fig 2 would be useful only of newly discovered markers are discussed.

On page 15, the authors state that “PDCD4 mediated PTX sensitivity in cancer cell lines and during adjuvant therapy of lung cancer patients with PTX, overall survival of patients correlated with high levels of PDCD4”. The readers need a brief background on PDCD4 to understand its impact in lung cancer disease dynamics. Kindly add a little about PDCD4

On page 14, authors talk about bTMB and ctDNA as novel markers. These should be subdivided in headings and more information on these markers should be given as these are new markers and have an impact on reader knowledge.

Author Response

(The authors gave the same response as above.)

Round 2

Reviewer 2 Report

The authors have complied with all suggested changes which has made the manuscript acceptable. Please go through the manuscript to check for spelling and Grammer mistakes. I would suggest you use a software such as Grammarly to help with english editing.